# Remembering to Be Fair Again: Reproducing Non-Markovian Fairness in Sequential Decision Making

**Domonkos Nagy**[*]                                    *domonkos.nagy2@student.uva.nl*
*Informatics Institute*
*University of Amsterdam*

**Lohithsai Yadala Chanchu**[*]                 *lohithsai.yadala.chanchu@student.uva.nl*
*Informatics Institute*
*University of Amsterdam*

**Kryštof Bobek**[*]                                    *krystof.bobek@student.uva.nl*
*Informatics Institute*
*University of Amsterdam*

**Xin Zhou**[*]                                         *xin.zhou@student.uva.nl*
*Informatics Institute*
*University of Amsterdam*

**Martin Smit**                                          *j.m.m.smit@uva.nl*
*Informatics Institute*
*University of Amsterdam*

**Reviewed on OpenReview:** *https://openreview.net/forum?id=H6DtMcZf5s*

## Abstract

Ensuring long-term fairness in sequential decision-making is a key challenge in machine learning. Alamdari et al. (2024) introduced *FairQCM*, a reinforcement learning algorithm that enforces fairness in non-Markovian settings via memory augmentations and counterfactual reasoning. We reproduce and extend their findings by validating their claims and introducing novel enhancements. We confirm that *FairQCM* outperforms standard baselines in fairness enforcement and sample efficiency across different environments. However, alternative fairness metrics (Egalitarian, Gini) yield mixed results, and counterfactual memories show limited impact on fairness improvement. Further, we introduce a realistic COVID-19 vaccine allocation environment based on SEIR, a popular compartmental model of epidemiology. To accommodate continuous action spaces, we develop *FairSCM*, which integrates counterfactual memories into a Soft Actor-Critic framework. Our results reinforce that counterfactual memories provide little fairness benefit and, in fact, hurt performance, especially in complex, dynamic settings. The original code, modified to be 70% more efficient, and our extensions are available on GitHub: `https://github.com/bozo22/remembering-to-be-fair-again`.

## 1 Introduction

As machine learning is increasingly involved in high-stakes decision-making tasks such as healthcare triaging (Sánchez-Salmerón et al., 2022) and loan approvals (Sheikh et al., 2020), it is becoming ever more critical to understand and address the risks of deploying such a system. One associated risk that directly affects end-users is that of fairness, in which a decision-maker may, intentionally or otherwise, benefit certain users over others. However, as shown in numerous studies (Liu et al., 2018; Hashimoto et al., 2018; Hu and Chen, 2018; D'Amour et al., 2020), actions that promote fairness in the short term can result in unexpected negative outcomes in the long term, which motivates the need to consider fairness in a non-Markovian light.

---

[*]Equal contribution.

Consider, for example, two countries that each receive an equal share of COVID-19 vaccines to promote fairness, allocated on a month by month basis. Even if they end up with the same amount of vaccines (which we would intuitively consider a fair allocation), it might be the case that one country received all its share at once, while the other only received it in small portions over time. Although the equal allocation appears fair in the long term, over time the outcomes diverge significantly, highlighting the need to consider fairness over the entire trajectory rather than in isolated, immediate decisions.

Sequential decision-making is the process of making decisions where each action influences future states and subsequent choices. Unlike static decision-making, where choices are made in isolation, sequential decision-making considers the long-term consequences of actions. Naive sequential decision-making systems face the challenge of decisions that can get stuck in feedback loops. These feedback loops can reinforce existing disparities, potentially creating a vicious cycle (Chohlas-Wood et al., 2023). An example of where such self-reinforcing loops can be found is in predictive policing systems (Ensign et al., 2018). These systems analyze crime data to forecast where future crimes are likely to occur, guiding law enforcement resource allocation. However, if the data used is biased—reflecting over-policing in certain communities—the system may perpetuate and worsen this bias. For instance, increased police presence in a neighborhood leads to more recorded offenses, which the system interprets as higher crime rates, prompting even more policing in that area. This creates a self-reinforcing cycle that disproportionately targets specific communities, exacerbating existing disparities.

These loops not only harm those directly affected in the short term, but also compound disadvantages over time. In recognition of these challenges, there is a growing ambition to leverage reinforcement learning to promote long-term fairness in algorithmic decision-making (Nashed et al., 2023) given its ability to balance immediate gains with desirable outcomes across a broader time horizon.

Past work that considers long-term fairness includes Jabbari et al. (2017), who examined approaches for adapting fairness principles within RL systems. They emphasized a dynamic and future-aware perspective on fairness, ensuring that fairness is preserved over the course of many decisions rather than focusing solely on individual actions in isolation. Sequential decision-making in Markovian settings has also been studied in Hu et al. (2024), which examines fairness in dynamic systems. These dynamic systems are modelled by a Markov decision process (MDP), where decisions are made sequentially, acknowledging that each decision can alter the underlying distribution of features or user behavior. However, Alamdari et al. (2024) claim that fairness assessments often depend on past actions taken during the decision-making process, rather than just the current state, rendering them inherently non-Markovian. To address this, they introduce the concept of non-Markovian fairness, exploring how memory and historical context can support the construction of fair policies in sequential decision-making settings. They propose *FairQCM*, a reinforcement learning approach designed to enhance fairness by leveraging *counterfactual experiences*, which allow the agent to learn how its actions would perform in hypothetical situations. While the results of the paper are promising, a number of the claims made paper lack experimental verification. Namely, while they claim that the method works in a wide variety of scenarios, motivated primarily on the COVID-19 vaccine roll-out problem mentioned above, their experiments do not include heterogeneous or dynamic stakeholder behavior, two key aspects of more realistic scenarios.

In this report, we first present and attempt to reproduce the main claims of (Alamdari et al., 2024), before conducting additional experiments that show how counterfactual experiences may negatively affect learning in environments with dynamic stakeholder behavior. In particular this report shows that:

- The ratio of counterfactual to real memories is an important hyperparameter affecting the performance of *FairQCM*.

- Counterfactual memories may actively hinder the learning of an effective policy in the case that the dynamics of the underlying system are more complicated. This is exemplified in our vaccine allocation scenario where the spread of an infection is modelled with an SEIR model (see section 3.2.4).

## 2  Scope of reproducibility

One of the main contributions of Alamdari et al. (2024), is their proposal of the Non-Markovian Fair Decision Process (**NMFDP**) framework. This allows standard algorithms for Markovian problems, such as Deep Q-Networks (DQNs) (Mnih et al., 2015), to be used for the non-Markovian problem of sequential fairness by extending the state with a "memory" that stores relevant information from preceding time steps.

They also introduce *FairQCM*, which uses DQN as a baseline but extends it with counterfactual memory augmentation. It uses the NMFDP framework to help generate long-term fair policies that consider past actions. Their work also introduces methods of expressing several different forms of fairness over time and explores the effect of memory augmentations on fairness performance.

We aim to reproduce the following claims made by Alamdari et al. (2024), which are tested using the **Resource Allocation** (Donut) (Katoh and Ibaraki, 1998) and **Simulated Lending** (Liu et al., 2018) environments utilizing their reproducibility package[1]:

**Claim 1:** Counterfactual reasoning improves fairness enforcement and sample efficiency.

**Claim 2:** Memory augmentations improve model performance significantly.

To investigate the first claim, we aim to reproduce the *Full* and *FairQCM* trials from the paper for both environments. We investigate if *FairQCM*, which uses counterfactual memories, performs better in both environments than the *Full* baseline (which consists of a regular DQN without counterfactual memories).

To further investigate the impact that counterfactual memories have on fairness performance, we test with varying balances of counterfactual memories to real memories (with a fixed memory buffer) to see what the optimal balance is.

To investigate the second claim, we replicate the trials for *FairQCM* and the other augmented memory baselines explored in the original paper to compare their results for both environments. We add a *No Memory* baseline to determine to what extent memory is important for the investigated tasks.

The original paper proposes vaccine allocation as an ideal real-world use case for *FairQCM* and its non-Markovian framework, although it did not evaluate it on environments that embodied the important traits of a realistic vaccine allocation environment. Such an environment is complex in the sense that it has changing dynamics sensitive to time and past actions. Additionally, *FairQCM* must demonstrate robustness across various fairness scoring functions, each of which offers distinct advantages and limitations that align with specific and potentially different fairness objectives users might have. The original paper tested *FairQCM* on environments with a discrete action space. A realistic COVID-19 vaccine allocation environment (and many other sequential decision-making tasks) requires a continuous action space, which would need a different off-policy learning algorithm. This leads naturally to the following extensions:

**Varying the aggregation reward function**  We test the fairness performance of *FairQCM* under different fairness scoring functions and compare it with various memory-augmented baselines. While many scoring functions were suggested in the original paper, only a limited number of them were actually tested. As such, it is not clear which fairness metrics can be effectively used by *FairQCM*.

**Introducing more complex stakeholder behavior:**  We introduce dynamic stakeholder behavior and dynamic behavior that depends on the agent's actions. This allows us to evaluate the original claims in scenarios with greater complexity and real-world accuracy.

**Assessing *FairQCM* in a realistic environment:**  We design a COVID-19 vaccine allocation environment according to the SEIR model (Tillett (1992)) with realistic hyperparameters for the COVID-19 virus, which allows modeling different populations, unique start states for each group, variable infection spread rates, and customized vaccine production schedules. We also propose a modified version of *FairQCM* called *FairSCM*, that works with continuous action spaces using the Soft Actor-Critic model.

---

[1]https://github.com/praal/remembering-to-be-fair

Table 1: Different types of memories stored by different baselines and the proposed *FairQCM*. $U(\tau_t)$ is a vector consisting of the utility values of each of the stakeholders, and $N$ denotes the number of stakeholders.

| Method | Way in Which Memory Is Represented |
|---|---|
| Full | $U(\tau_t)$ |
| Min | $U(\tau_t) - \min_i U(\tau_t)_i$ |
| RNN | Instead of storing memories directly, it uses a recurrent Q-network to remember relevant information from past actions. |
| Reset | Stores $U(\tau_t)$, and whenever the utility values are all equal, resets them to 0. |
| FairQCM | $\begin{cases} U(\tau_t), & \text{in resource allocation environment} \\ U(\tau_t) - \min_i U(\tau_t)_i, & \text{in simulated lending environment} \end{cases}$ |

## 3 Methodology

### 3.1 Methodology reproducing original paper

The replication package of the original paper was used to reproduce its experiments, in which the proposed method, *FairQCM* and four baselines were tested: *Full*, *Min*, *Reset* and *RNN*. Besides the above five methods, we also added a *No-Memory* baseline.

#### 3.1.1 Introduction to the proposed method and baselines

The proposed method, *Fair Q-Learning with Counterfactual Memories* (*FairQCM*) is based on Deep Q Networks (DQN) Mnih et al. (2015). Its main contribution is that instead of feeding **states** of the environment into the Q-network, *FairQCM* uses **memory-augmented states**. This augmentation converts a non-Markovian fairness problem into a Markovian one by keeping track of information related to past actions. For example, if the agent's goal is to distribute goods between different stakeholders, it could store the amount of goods it has given to each stakeholder up to the current time step in the memory.

At time step $t$ in trajectory $\tau$, the *stakeholder status* $U(\tau_t)$ is a vector consisting of the utility values of each of the stakeholders. The differences between the proposed baselines are the types of memory they store, the details of which are found in Table 1. Notably the RNN considers actions and memories in an end-to-end fashion, as memories are not stored explicitly, but are implicitly represented in the Q-network itself.

Although the original paper explored how different memory representations affect performance, they did not compare the performance of the various memory-augmented baselines to a classic memory-less baseline, which does not extend state with a memory. This is a crucial part of the evidence to suggest that memory is needed to perform well at the examined tasks of resource allocation and lending, and gives us more insight into how memory augmentations improve performance. This is why we added the *No-Memory* baseline, where the DQN receives the state and no memory.

We also investigated how performance depended on the composition of the replay buffer in terms of regular memories, those experienced by the agent during training, and counterfactual memories, those constructed algorithmically to aid learning (see Section 3.2.1). By analyzing different compositions of the replay buffer, we can assess whether an optimal ratio exists that maximizes learning efficiency while maintaining fairness and stability. This investigation provides insights into the role of counterfactual experiences in shaping policy updates and whether their inclusion leads to more robust generalization.

#### 3.1.2 Resource allocation environment

The donut allocation experiment has the setup of $n = 5$ customers, with customer $i$ being in front of the counter with constant probability $p_i = 0.8$ at each time step $t$. The donut shop bakes one donut at each time step, and the server chooses one customer to give out the donut to. If the server chooses a customer that is not at the counter, the donut is wasted. The fairness scoring function $W$ serves as the reward function. In

the original paper, the reward is based on the *Nash* welfare score as shown in Equation 1.

$$w_t = W(U(\tau_t)) = \log(\text{Nash}(U(\tau_t) + 1)) = \sum_{i=1}^{n} \log(U(\tau_t)_i + 1), \tag{1}$$

where $U(\tau_t)_i$ is the number of donuts allocated to customer $i$ so far at time step $t$. The agent's goal then is to allocate donuts in a fair manner across customers, as measured by the *Nash* welfare score.

We averaged the results over 10 experiments, each consisting of 500 episodes of 100 time steps. We used the same training hyperparameters as the original paper, which are detailed in Appendix A.2.

### 3.1.3 Simulated lending environment

In this experiment Alamdari et al. (2024) adopt the lending environment proposed by Liu et al. (2018) as a version of the established fair machine learning testbed, consumer lending Dwork et al. (2012); Hardt et al. (2016). The original experiment examines four loan applicants divided into two protected groups ($A$ and $B$), with each applicant characterized by a credit score $C$ that represents the probability of loan repayment. The credit score distribution varies between these two groups. At each step, a subset of applicants applies for a loan, and the agent (bank) determines which applicant to approve. Each applicant $i$ applies with a fixed probability of $p_i = 0.9$. Successful loan repayment increases the bank's utility by $r$ and the applicant's credit score by $c$, while defaulting decreases both. States include the subset of applicants applying for loans, their credit scores, and the cumulative profit margin.

Fairness in Alamdari et al. (2024) is assessed per timestep using the *Relaxed Demographic Parity (DP) Score*:

$$W(U(\tau_t)) = -\left| \sum_{i \in A} U(\tau_t)_i - \sum_{i \in B} U(\tau_t)_i \right|, \tag{2}$$

where $U(\tau_t)_i$ represents the cumulative number of loans granted to applicant $i$ up to time step $t$. The agent's goal is to maximize the discounted sum of $W \circ U$, subject to two constraints. A significant negative reward is imposed if at least 10% profit margin is not achieved by the end of an episode, and a minor penalty is applied for granting loans to non-applicants.

The following baselines are used for the evaluation of the experiment: *Full*, *Min*, and *RNN*. The *Min* memory is also utilized by *FairQCM*.

## 3.2 Methodology beyond original paper

The original paper proposes vaccine allocation as a real-world use case for *FairQCM* but did not test it in environments that reflect the complexities of sequential decision-making, where stakeholder behavior changes over time. Additionally, *FairQCM* should be robust across various fairness scoring functions, each suited to different fairness objectives. We aim to explore the performance of *FairQCM* on different fairness scoring functions and dynamic stakeholder behavior. Furthermore, *FairQCM* is limited by the fact that it can only work with discrete action spaces, whereas many realistic sequential decision making tasks (such as vaccine allocation) are best modeled using a continuous action space, and consequently, a different off-policy learning approach, such as Soft Actor-Critic (Haarnoja et al., 2018).

### 3.2.1 Evaluating the impact of counterfactual memories on fairness

To further investigate the impact that counterfactual memories have on fairness performance, we test with varying balances of counterfactual memories to real memories (with a fixed memory buffer) to see what the optimal balance is. In the Donut environment with 5 stakeholders, we analyzed how FairQCM's performance varies with the number of counterfactual memories stored in a fixed replay buffer of size 6,400. During training, FairQCM samples from this buffer, which we populate with different amounts of counterfactuals per time step. We define $m$ as the number of maximum additional donuts each stakeholder could receive, ranging from 0 to 4. For instance, with $m = 2$, each stakeholder could have 0, 1, or 2 additional donuts, yielding $3^5 = 243$ counterfactual states. We tested $m = [0, 1, 2, 3, 4]$, corresponding to 0, 1, 32, 243, and 1,024 counterfactuals per time step.

### 3.2.2 Exploring different fairness scoring functions

In the original paper, only the *Nash* welfare score was applied as a reward function. In order to verify the performance of *FairQCM* using different definitions of fairness, we use the *Egalitarian* and *Gini* welfare scores as reward functions to train separate models from scratch for the Donut environment. Compared to *Nash*, these welfare scores emphasize demographic parity with a decreased focus on achieving high overall utilities.

The *Egalitarian* reward function (as shown in Equation 3) aims to minimize the differences from the mean, that is, equal the number of donuts allocated to each customer.

$$w_t = W(U(\tau_t)) = -\sum_{i=1}^{n} |U(\tau_t)_i - \overline{U(\tau_t)}|. \tag{3}$$

The *Gini* reward function at time step $t$ is defined in Equation 4.

$$w_t = 1 - G, \text{ where } G = \frac{\sum_{i=1}^{n}\sum_{j=1}^{n} |U(\tau_t)_i - U(\tau_t)_j|}{2n^2\overline{U(\tau_t)}}. \tag{4}$$

$G$ is a measure of inequity, defined as the mean of absolute differences between all pairs of individuals for some measure, which is $U(\tau_t)_i, i \in \{1, 2, ..., n\}$ in our case. The minimum value of $G$ is zero when all the $U(\tau_t)_i, i \in \{1, 2, ..., n\}$ are equal. The theoretical maximum is one for an infinitely large set of individuals where all $U(\tau_t)_i, i \in \{1, 2, ..., n\}$ is zero except one individual (ultimate inequality).

Besides, we also introduce the *Rawlsian* reward function as below, which is the reward function of COVID-19 allocation simulation environment. It aims to maximize the minimum number of the utility:

$$w_t = W(U(\tau_t)) = min(U(\tau_t)). \tag{5}$$

We applied the *Egalitarian* and *Gini* fairness scores to train and evaluate *FairQCM* along with four baselines: *Full*, *Min*, *Reset* and *RNN*. We averaged the results over 10 experiments, each consisting of 1000 episodes of 100 time steps. We used the same training hyperparameters as the original paper which can be found in the Appendix A.2.

### 3.2.3 Adding dynamic stakeholder behavior

The original paper considers customer appearance probabilities that are both constant over time and uniform across customers. To better simulate real-world variability, we introduce two new variants in the donut allocation example.

In the first experiment, we assign distinct, static appearance probabilities to each customer: $p = [0.6, 0.7, 0.8, 0.9, 1.0]$. These values are chosen arbitrarily and do not change throughout the episode.

In the second experiment, we introduce an action-dependent probability for customer appearance. Initially, each customer has a probability of $p = 0.6$ of appearing. However, when a customer receives a donut, their probability of coming in the subsequent round increases by 0.1. Conversely, if a customer does not receive a donut for two consecutive iterations, their appearance probability resets to 0.6. This mechanism allows customer appearance dynamics to be influenced by past allocations, introducing simple agent-driven nonstationarity.

### 3.2.4 COVID-19 allocation simulation

To investigate the applicability and effectiveness of counterfactual memory augmentations in addressing complex, real-world problems, we created a COVID vaccine allocation gym environment. In this scenario, supply constraints necessitate that allocation decisions be made over time, with each decision impacting future outcomes. For example, prioritizing a specific demographic in one period can influence overall population immunity or fairness in subsequent periods. As the vaccine allocation strategy greatly affects all infected regions, this problem serves as a perfect testbed to compare different fair learning methods.

Furthermore, vaccine allocation problems are often modeled as multi-stakeholder systems, where regions, hospitals, or demographics act as agents competing for limited resources (Li and Huang, 2022; Yarahmadi et al., 2023; Dalgıç et al., 2017). The ability to integrate fairness into such multi-agent systems provides a robust framework for tackling these challenges. By combining previous extensions of our experiments, such as the behavior under different aggregation or reward functions and the adaptability to dynamic conditions, we found it to be naturally suited to study the task of vaccine allocation.

We developed our vaccine allocation environment to reflect both the complexity of disease spread and the challenges of distributing limited resources fairly. At its core is the well-known SEIR model (Taghizadeh and Mohammad-Djafari, 2022), an epidemiological framework that categorizes individuals according to their disease status: **Susceptible** (S) for those who can contract the disease, **Exposed** (E) for those infected but not yet infectious, **Infected** (I) for those actively transmitting the disease, and **Recovered** (R) for those who have either gained immunity or passed away. The model captures how individuals transition among these compartments over time as governed by Equation 6.

$$\frac{dS}{dt} = -\beta \cdot S \cdot \frac{I}{N}, \quad \frac{dE}{dt} = \beta \cdot S \cdot \frac{I}{N} - \sigma \cdot E, \quad \frac{dI}{dt} = \sigma \cdot E - \gamma \cdot I, \quad \frac{dR}{dt} = \gamma \cdot I, \tag{6}$$

where $N$ is the total population size; $\beta$ is the transmission rate; $\sigma$ is the rate of progression from exposed to infectious ($1/\sigma$ is the incubation period); $\gamma$ is the recovery rate ($1/\gamma$ is the infectious period).

Our environment models multiple regions with distinct populations, infection dynamics, and initial states (e.g., exposed or infected counts). It reflects real-world disparities, where infection rates and susceptibility vary by demographics and infrastructure. Additionally, vaccine production fluctuates over time, adding complexity to allocation decisions. To address this, we implement a customizable vaccine production schedule, requiring the allocation agent to balance infection control and fairness under dynamic supply conditions.

We define the stakeholder utility for region $i$ at time step $t$ as:

$$U(\tau_t)_i = 0.04(V(\tau_t)_i - P(\tau_t)_i) - E(\tau_t)_i, \tag{7}$$

where $V(\tau_t)_i$ is the percentage of allocated vaccines up to time step $t$ that were given to region $i$, $P(\tau_t)_i$ is the percentage of the overall population that lives in region $i$, and $E(\tau_t)_i$ is the percentage of the overall population in region $i$ that got exposed to the virus at time step $t$.

Intuitively, we can assume that each region has two main objectives: they want to minimize the number of inhabitants that become exposed to the virus (hence $-E(\tau_t)_i$), while also making sure that the percentage of vaccines they receive is not disproportionately smaller than the percentage of the overall population they account for (hence $V(\tau_t)_i - P(\tau_t)_i$). We can further assume that the main focus of each region is to reduce the number of exposed inhabitants (as it has a direct effect on the spread of disease), while maintaining a fair proportion of received vaccines can be thought of as a regularizer, hence the scaling factor 0.04. The reward at each time step is the Rawlsian welfare score. The utility and aggregation functions serve as feasible fairness baselines, and are by no means attempts at establishing an objective measure of a fair vaccine allocation.

As our environment has continuous actions, using DQN would only be possible with severe limitations, such as quantizing the action space. To avoid such limitations, we opted to use Soft Actor-Critic (SAC) (Haarnoja et al., 2018) instead, which is an off-policy learning method with continuous actions. Similar to FairQCM, we refer to the memory-augmented version of SAC as FairSCM (Fair SAC with Counterfactual Memories).

### 3.3 Computational results

The reproducibility of the results was based on the original code repository. We ran experiments using an AMD Ryzen 2600 CPU and a Nvidia RTX 3060 GPU. This original code took around 24 hours to run all the experiments. We significantly improved the efficiency and reduced the time to run all experiments to 7 hours, which is a speedup of around 70%. Running all experiments (reproducing original claims as well as running our extensions) took less than 12 hours in total. This code is available on GitHub[2]. The speedup was partly achieved through modifying the code to take advantage of a GPU.

---

[2]https://github.com/bozo22/remembering-to-be-fair-again

# 4 Results

We were able to successfully reproduce the original experiments, running *FairQCM* and the other baselines on both the resource allocation and the simulated lending environments. In addition to replicating the original findings, we extended the experiments to examine how *FairQCM* performs under alternative fairness metrics and within dynamic environments.

We evaluated different fairness scoring functions beyond those initially tested. We also explored *FairQCM*'s adaptability in environments where conditions change over time, specifically focusing on scenarios where stakeholder probabilities evolve dynamically.

## 4.1 Results reproducing original paper

### 4.1.1 Resource allocation

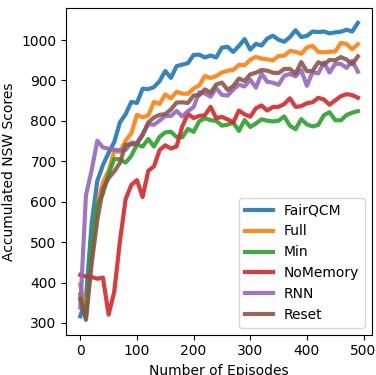 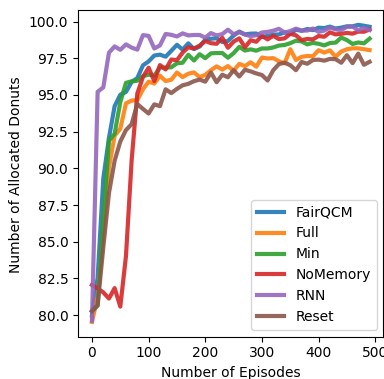

Figure 1: Replication of the resource allocation experiment from Alamdari et al. (2024) with the additional *No-Memory* baseline. Results correspond with the original paper.

The replication results shown in Figure 1 align with the results in the original paper. They support the claims that *FairQCM* outperforms other baselines and learns to allocate donuts effectively. Similarly to the original paper, *Min* has the least Nash welfare score, and *RNN* adapts faster than others to learn to allocate the donuts effectively.

**Claim 1:** Our results confirm that *FairQCM* outperforms the baselines, particularly in the resource allocation environment, where it achieves higher fairness and faster learning. Compared to *Full*, *FairQCM* demonstrates greater sample efficiency, requiring fewer steps for improved performance.

**Claim 2:** Our second focus was on memory augmentations and their significant impact on model performance. The results in both environments validated that different memory structures, such as *Full*, *Reset*, and *RNN*, influence fairness outcomes in distinct ways. We see the same ordering in the performances of the different memory augmentations as in the original paper.

### 4.1.2 Simulated lending

Our replication of the experiment verifies key claims made by the original paper. All baseline methods achieve the desired profit margin, supporting the validity of the experimental setup (Figure 2).

**Claim 1:** This claim holds in our replication. *FairQCM*, which enhances the *Min* baseline with counterfactual experiences, achieves a higher fairness score than *Min* alone. It also demonstrates higher sample efficiency, improving more rapidly than the other tested baselines.

**Claim 2:** Our findings support this claim. All memory-augmented baselines outperform both the *RNN* and *No-Memory* baselines by a substantial margin.

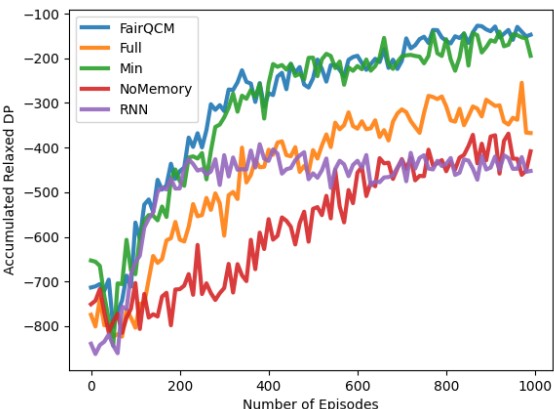

Figure 2: Replication of the simulated lending experiment from Alamdari et al. (2024), incorporating an additional *No-Memory* baseline. The results align with the original findings, reinforcing their conclusions.

However, we observed that the *Full* and *Min* baselines performed better in our replication than reported in the original study. This discrepancy occurred despite using the provided replication package, the same seed, and the exact command sequence described by the authors.

## 4.2 Results beyond original paper

### 4.2.1 Effect of counterfactual memories on fairness

Although results from the original paper suggest that counterfactual memories improve fairness performance, further analysis of counterfactual balance experiments (Figure 3) suggests that the specific proportion of counterfactual and real memories in the replay buffer influences fairness performance significantly as well: too many counterfactual memories perform worse than no counterfactual memory augmentation present. This might be due to the increasing proportion of unrealistic counterfactual states that increase with $n$. With higher values for $n$, more states that would normally never appear in a trajectory would be produced as counterfactual states, that the DQN learns from. These additional "impossible" counterfactual states add noise to the learning process. For low values of $n$, the additional benefit of increased (realistic) counterfactual memories seems to outweigh the performance degradation caused by the added noise (of the unrealistic counterfactual memories).

### 4.2.2 Results of different fairness scoring functions

In Figure 4a, *FairQCM* and four other baselines—*Full*, *RNN*, *Reset*, and *Min*—are trained using the *Egalitarian* welfare score as the reward function and evaluated with the same score as the metric. In Figure 4b, all five models are trained and evaluated using the *Gini* welfare score. As shown in Figure 4a, all baselines except *FairQCM* tend to give out fewer donuts in total as the training progresses. This is likely due to the fact that the *Egalitarian* welfare score considers only the fairness of the allocation but not the overall amount of stakeholder utilities, so wasting donuts does not decrease the reward. *RNN* achieves highest accumulated *Egalitarian* scores but this is because it learns to give out least amount of donuts. In contrast, *FairQCM* tends not to waste more donuts as it learns fair allocation policy and thus performs better than other baselines. However, Figure 4b shows that *FairQCM* underperforms *Full*, *Min* and *Reset* both in terms of welfare score and number of allocated donuts under the evaluations of *Gini* welfare score. Based on these results, *FairQCM* is less "fair" across stakeholders and also produces less utility since it tends to waste more donuts compared to other baselines.

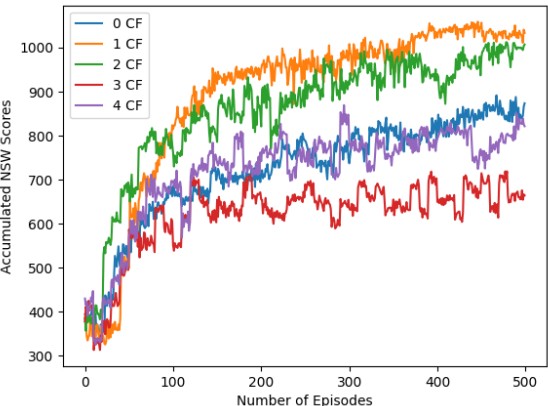

Figure 3: FairQCM's performance in the Donut environment (5 stakeholders) as a function of counterfactual memories in a fixed 6400-memory replay buffer. As $n \in \{0, 1, 2, 3, 4\}$ increases, the number of counterfactual states grows exponentially. Too few counterfactuals limit learning, while too many overwhelm the buffer, indicating an optimal range.

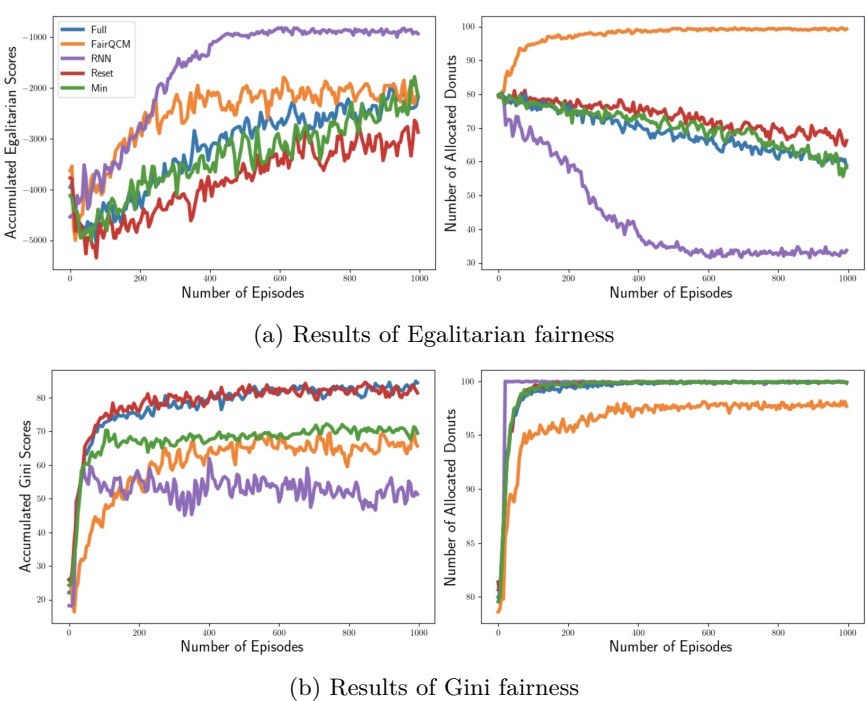

(a) Results of Egalitarian fairness

(b) Results of Gini fairness

Figure 4: *FairQCM* underperforms *RNN* (top left) but learns to allocate donuts effectively (top right) under the guidance and assessment of *Egalitarian* welfare score; *FairQCM* underperforms *Full*, *Min* and *Reset* (bottom left) and wastes more donuts under the guidance and assessment of *Gini* welfare score.

Overall, *FairQCM* performs worse under the evaluations of the *Gini* welfare score compared to the *Egalitarian* and *Nash* welfare scores. It is an interesting question for future work how well *FairQCM* can generalize to multiple scoring functions simultaneously.

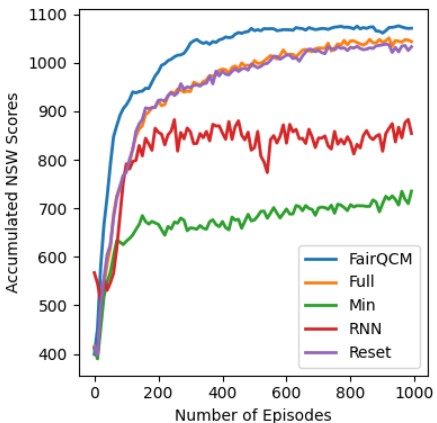 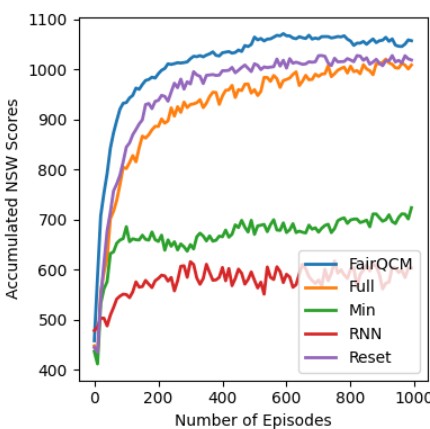

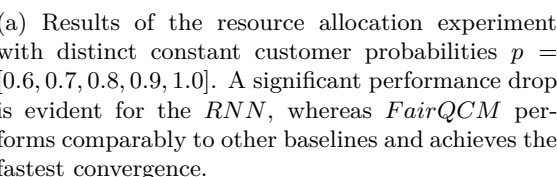

(a) Results of the resource allocation experiment with distinct constant customer probabilities $p = [0.6, 0.7, 0.8, 0.9, 1.0]$. A significant performance drop is evident for the $RNN$, whereas $FairQCM$ performs comparably to other baselines and achieves the fastest convergence.

(b) Resource allocation experiment with dynamically changing customer probabilities, illustrating the underperformance of $RNN$. The initial probability of customers coming is set to $p = 0.6$, increasing by 0.1 for the customer receiving a donut, resetting to the original value after 2 steps without donut allocation.

Figure 5: Results of extended resource allocation experiments

### 4.2.3 Dynamic stakeholder behavior

The results reveal a significant performance drop in the $RNN$ and $Min$ baselines, which struggle to adapt to the increased complexity of the environment for both distinct constant (Figure 5a) and dynamically changing probabilities (Figure 5b). In contrast, the $FairQCM$ approach confirms its fast convergence and sample efficiency across both experimental settings, even achieving superior results in the action-dependent environment.

### 4.2.4 Evaluation of FairSCM under the COVID-19 vaccine allocation simulation

Our results show that all baselines were able to successfully optimize for our dual reward function. As Figure 6 demonstrates, both the overall number of infections and the inequality in vaccine allocations decrease as the models learn, while the cumulative $Rawlsian$ welfare scores increase. Thus, the models learn a policy that distributes vaccines fairly between regions while also controlling the spread of the virus effectively.

The results also demonstrate that counterfactual memory augmentations for this complex environment have a negative effect on performance. The policies that FairSCM learns underperform other baselines both in terms of fairness (cumulative Rawlsian welfare) and effective disease control (number of infections across all regions). We hypothesize that the reason behind this decrease in performance is a result of filling the buffer with noisy counterfactual transitions. For simpler problems such as donut allocation, the model can learn the environment dynamics easily, and the counterfactual transitions can provide additional robustness by showing the model transitions that might be unlikely under its policy. However, for more complex environments such as our COVID-19 simulation, the model already has a difficult time figuring out the dynamics of the environment, so an extensive amount of improbable counterfactual transitions can effectively act as noise, hindering the model's ability to understand the true dynamics of the environment.

## 5 Discussion

Based on the results reproducing the original paper, we verified the two main claims which highlight the important roles of counterfactual reasoning and memory augmentations. However, we showed that these contributions do not generalize well to more complex environments and different fairness scoring functions.

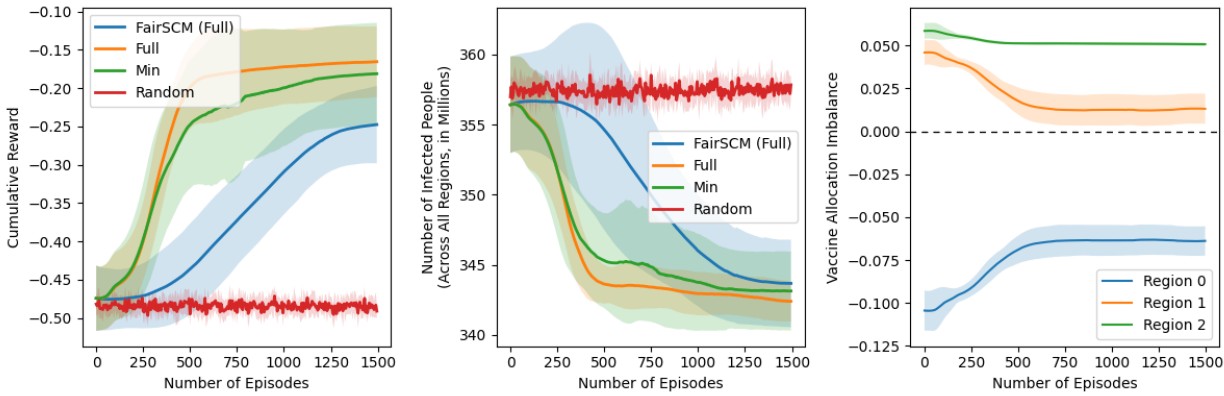

Figure 6: Results of the vaccine allocation experiment with the mean values plotted and the standard deviations shown as shaded areas around the mean. The left plot shows the cumulative rewards, while the center plot shows the overall number of people who get infected during the simulation. The right plot shows the vaccine allocation imbalance (difference between percentage of allocated vaccines and ratio of overall population) for each of the three regions for the *Full* baseline.

The performance of *FairQCM* under the evaluation of the *Gini* welfare score suggests that counterfactual memories do not always improve fairness. Furthermore, our results in the COVID-19 environment show that counterfactual memories for complex environments can have a negative effect on the model's fairness performance.

The replication of the original results is not difficult to obtain as the authors provided a sufficiently working codebase, however their implementation only supports training on the CPU, which is inefficient. In addition, the codebase contains many pieces of redundant code, as the author copies classes and functions for different environments (resource allocation and simulated lending). We made additional efforts to aggregate all the source files and extend them to enable the training on GPU. Furthermore, we found some mathematical definitions, design choices and surprising results in the original paper somewhat difficult to understand. We sent our questions to the authors, but have not received a response by the time of submission.

**Environemntal Impact:** Our optimizations of the original codebase yielded a 70% speedup, reducing the energy consumption to a similar degree. While the reproducibility experiments took only 7 hours to run, running and fine-tuning our extensions (including re-running experiments for testing and hyperparameter optimization) took approximately 80 hours in total. Experiments were conducted using private infrastructure, with an estimated carbon efficiency of approximately 0.22 $kgCO_2eq$/kWh (average global energy efficiency from Ritchie et al. (2020)). Overall, 87 hours of computation was performed on an NVIDIA RTX 3060 GPU (170W TDP). Total emissions are estimated to be 3.2538 $kgCO_2eq$. To put this into perspective, a single cow produces an estimated 8.2-11 $kgCO_2eq$ per day (Garcia (2024)).

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

# A   Appendix

## A.1   Implementation details for the COVID-19 simulation experiments

**State Representation**   The (unnormalized) state space of our environment consists of the number of susceptible, exposed, infected and recovered individuals for each region, as well as the number of vaccines available for distribution at the current time step. These values are scaled with the overall population.

**Memory Representation**   The (unnormalized) memory for the *Full* baseline at each time step consists of the number of vaccines distributed so far to each of the regions. The *Min* baseline uses a similar representation, but subtracts the minimum value. These values are scaled with the overall number of vaccines distributed so far. *FairSCM* uses the same memory as the *Full* baseline.

**Actions**   The action space is made up of continuous values between -1 and 1 for each of the regions. The softmax function is applied to the actions to determine the percentage of available vaccines to distribute to each of the regions.

**Environment Dynamics** For our experiments, we used three regions with an overall population of one billion. The SEIR parameters for the regions were determined by choosing the parameters estimates of three random countries from Sharma et al. (2022). The region populations, as well as the initial ratios of susceptible, exposed and infected individuals were determined arbitrarily with a focus on diversity between regions.

**Counterfactual Memories** We constructed counterfactual memory augmentations by randomly shifting the values in the unnormalized memory (number of vaccines given to each of the regions). We sampled the perturbations uniformly between -10,000,000 and 10,000,000. For each real transition, we added three counterfactual transitions to the buffer.

**Neural Network Architectures** Our SAC agent uses identical actor and critic architectures, consisting of three fully-connected layers of 64, 32 and 16 units with ReLU activation functions. We used the implementation provided by Stable-Baselines3 (Raffin et al. (2021)), and extended it with our own modifications.

**Hyperparameters** See Table 2.

Table 2: COVID-19 Simulation Hyperparameters

| Hyperparameter | Full and Min approaches | FairSCM |
|---|---|---|
| Episode Length | 24 | 24 |
| Learning Rate | 0.00003 | 0.00003 |
| Discount Factor ($\gamma$) | 0.99 | 0.99 |
| Replay Buffer Size | 5000 | 15000 |
| Batch Size | 128 | 512 |
| Soft Update Coefficient ($\tau$) | 0.000001 | 0.000001 |
| Entropy regularization coefficient | 0 | 0 |

### A.2 Implementation details for resource allocation experiments

**Hyperparameters** See Table 3.

Table 3: Resource Allocation Hyperparameters

| Hyperparameter | Full, Min, and Reset approaches | RNN approach | FairQCM |
|---|---|---|---|
| Episode Length | 100 | 100 | 100 |
| Learning Rate | 0.0001 | 0.002 | 0.0001 |
| Discount Factor ($\gamma$) | 0.95 | 0.95 | 0.95 |
| Min Exploration Rate ($\epsilon$) | 0.2 | 0.2 | 0.2 |
| Replay Buffer Size | 400 | 1000 | 6400 |
| Batch Size | 64 | 256 | 2048 |

### A.3 Additional results of exploring different fairness scoring functions

Besides the *Egalitarian* and *Gini* welfare scores, we also conducted experiments on the *Utilitarian* and *Rawlsian* welfare scores.

The *Utilitarian* reward function simply takes the sum of the numbers of donuts allocated to each customer so far, and aims to maximize the overall number of donuts given out:

$$w_t = W(U(\tau_t)) = \sum_{i=1}^{n} U(\tau_t)_i. \tag{8}$$

**Results based on the *Utilitarian* welfare score:** As shown in Figure 7, after 200 episodes the *Utilitarian* score of *FairQCM* starts to decrease and the number of donuts given out also starts to decrease. It is surprising as the *Utilitarian* welfare score aims to maximize the total number of donuts given out.

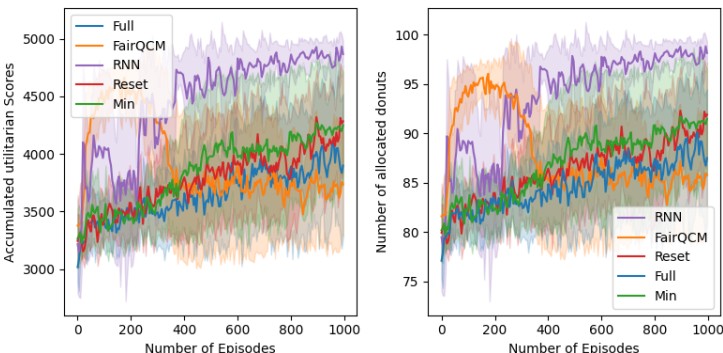

Figure 7: The *Utilitarian* score of *FairQCM* surprisingly starts to decrease after 200 episodes.

**Results based on the *Rawlsian* welfare score:** As shown in Figure 8, it can be found that *Min* has zero *Rawlsian* welfare score and hardly allocates any donut to customers. Besides, *FairQCM* underperforms *Full* and *Reset* baselines.

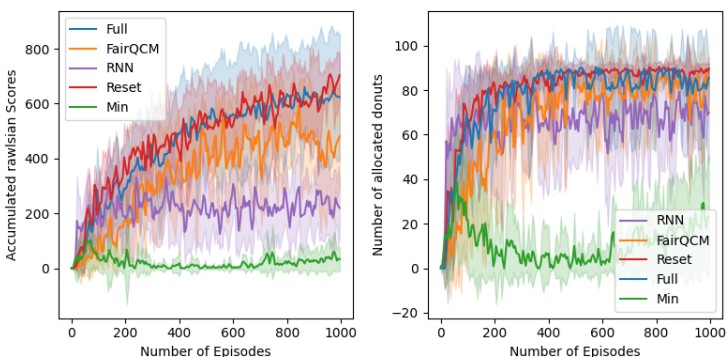

Figure 8: *Min* performs poorly under the assessment of *Rawlsian* welfare score.

### A.4 Additional results of dynamic stakeholder probabilities

**Heterogeneous probabilities (action-independent):** In this experiment, we introduce time-dependent probabilities, allowing customer appearance likelihoods to vary dynamically at each time step. This behavior is modeled using a sigmoid-like function:

$$f(t) = \frac{1}{1 + e^{-\text{steepness} \cdot (t - t_{\text{mid}})}} \tag{9}$$

where $t$ is the current time step, and steepness and $t_{\text{mid}}$ are customer-specific hyperparameters. Here, $t_{\text{mid}}$ denotes the time step at which the probability reaches 0.5, while steepness controls the transition rate, shaping the curve's gradient.

The results indicated an unexpected increase in the variance of *FairQCM* after approximately 500 episodes, which remains unexplained (Figure 9). Additionally, the *RNN*'s subpar performance observed in this experiment was consistent with our findings from previous experiments.

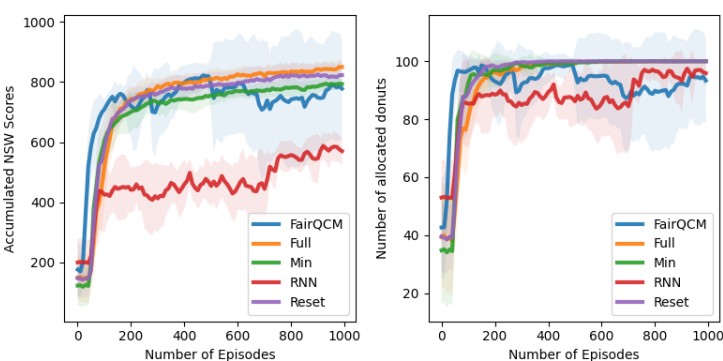

Figure 9: Resource allocation experiment with dynamically changing customer probabilities, illustrating the subpar performance of $RNN$ and an unexpected increase in variance for $FairQCM$. The probability modeling hyperparameters used in this experiment were: $t_{\mathrm{mid}} = [50, 50, 50, 75, 25]$ and steepness $= [0.9, -0.9, 0.1, 0.6, 0.5]$.

