# OpenReview forum: "Remembering to Be Fair Again: Reproducing Non-Markovian Fairness in Sequential Decision Making"
_TMLR — Accepted by TMLR_

### Review · Reviewer_NiT1 · 2025-03-07

**Summary Of Contributions:**

This paper provides additional empirical evaluation for the FairQCM method introduced by  Alamdari et al. (2024). FairQCM is an DQN algorithm that sees environment states augmented with past actions (non-Markovian) and its goal is to enforce fairness constraints in the context of sequential decision making. The new experiments extend the existing evaluation along several axis including: additional baselines; more realistic settings with continuous action space; robustness over different fairness definitions; the impact of counterfactual memories on fairness.

**Audience:**

Yes

**Broader Impact Concerns:**

I do not have any such concerns.

**Claims And Evidence:**

Yes

**Requested Changes:**

Please find below my suggestions on how to improve the paper. I am already positive about this paper and none of the following recommendations are "mandatory" from my side. That said, I would be greatful to the authors if they take my comments into consideration and implement whatever they might find it could help improve the quality of the paper.

- Even thought the paper covers well the existing work it is based on, I would appreciate a more clean and concentrated explanation on why additional experiments are needed and what past evaluation lacked. This should be added in the Introduction section, in order to highlight the gap that the current paper tries to fill.

- I know that FairQCM is derived from prior work, but it is not motivated in the current one. In other words, the authors should mention why they believe that this is an appropriate approach to enforcing fairness in sequential decision making.

- A more clear explanation of the distinction between regular and counterfactual memories on 3.1.1 is needed.

- The exploration of additional RL algorithms for both the discrete and continuous space cases would significantly strengthen empirical results.

- The results on the robustness over different fairness functions are disheartening. I appreciate that the authors included these results and did not hide them. However, I believe that this motivates a more extended investigation, i.e., experiments with additional fairness functions. Furthermore, the authors should include additional discussion about these results.

- Regarding related work section, papers that study notions related to fairness in sequential decision making could be also covered for generality. For example, works that explore notions such as interpretability, accountability, harm etc.

**Strengths And Weaknesses:**

**Strengths**

The paper is well-structured, with clearly stated motivation and contributions. It does not oversell and also covers appropriately related and previous works on the topic of fairness in sequential decision making. The additional experiments are interesting and reasonable. The most important contribution in my opinion is the newly proposed COVID environment. I believe that it could aid research on RL and causal RL which lacks simulation testbeds on healthcare, especially in the continuous action space. Given the success that similar environments, e.g., Sepsis, had over the past years I am optimistic that this testbed can have an impact.

**Weaknesses**

Arguably, this paper has very limited novelty as it simply extends the experimental section of an existing paper. However, in my opinion this is fine, and does not diminish the importance of the contribution. I have several suggestions on how to improve the paper, please see my comments on the **Requested Changes** section.

---

> ### Author Response · Authors · 2025-03-18
> **Response to your review**
>
> We thank the reviewer for their insightful comments and constructive feedback. Below, we address the questions and concerns raised by the reviewer.
>
> - Even though the paper covers well the existing work it is based on, I would appreciate a more clean and concentrated explanation on why additional experiments are needed and what past evaluation lacked. This should be added in the Introduction section, in order to highlight the gap that the current paper tries to fill.
>
> We’ve clarified why the experiments were necessary and the gap in existing literature.
>
> “a number of the claims made by the paper lack experimental verification. Namely, while they claim that the method works in a wide variety of scenarios, motivated primarily on the COVID-19 vaccine rollout problem mentioned above, their experiments do not include heterogeneous or dynamic stakeholder behavior, two key aspects of more realistic scenarios.”
>
> - The exploration of additional RL algorithms for both the discrete and continuous space cases would significantly strengthen empirical results.
>
> We agree. However, a comprehensive exploration of additional algorithms would exceed the scope of this paper. To maintain focus, we adhere to the original algorithms and their natural extensions, such as the soft actor-critic. Nonetheless, we are eager to explore these alternatives in future work.
>
> - A more clear explanation of the distinction between regular and counterfactual memories on 3.1.1 is needed.
>
> We’ve clarified this and cross linked to section 3.2.1, which explains this in more detail.
>
> - The results on the robustness over different fairness functions are disheartening. I appreciate that the authors included these results and did not hide them. However, I believe that this motivates a more extended investigation, i.e., experiments with additional fairness functions. Furthermore, the authors should include additional discussion about these results.
>
> We restructure section 4.2.2 to do a more comparative discussion about the results of the different fairness functions. For example, the underperformance of FairQCM on the Gini score compared to the Egalitarian score.

---

> > ### Comment · Reviewer_NiT1 · 2025-03-20
> > **Thank you**
> >
> > Thank you for your answer.

---

### Review · Reviewer_pp2W · 2025-03-11

**Summary Of Contributions:**

- The paper reproduces the results of Alamdari et al. (2024) where the method FairQCM is introduced and evaluated on resource allocation and a simulated lending problem.
- The paper extends the results by evaluating the resource allocation problem on varying amount of counterfactual memories in the replay buffer.
- The paper extends the results by comparing different welfare scores for the resource allocation problem.
- The paper extends the results by also using dynamic stakeholder behavior.
- The paper adjust the method FairQCM to FairSCM, the SAC equivalent of the original method.
- The paper evaluates FairSCM on a complex environment.

**Audience:**

Yes

**Broader Impact Concerns:**

I have no critique with respect to broader impact concerns.

**Claims And Evidence:**

Yes

**Requested Changes:**

- It is not clear at the end of Section 2 what the reference is for.

- Can you elaborate why in Table 1, method RNN, it translates the RNN to a recurrent **Q**-network? What makes the RNN a Q-network here?

- Please add a short sentence in the resource allocation environment (3.1.2) to specify the desired goal.

- I would strongly encourage cross-linking throughout the paper. The SEIR model is explained in 3.2.4, but already referenced several times before in the paper. In my opinion, it strengthens the comprehensibility if a link / hint is given where the explanation can be found since it is explained in a later part in the paper.

- If an equation is referenced explicitly like "(as shown in equation 3)", *E*quation should be uppercased.

- The used line for mean should span across the whole term of which the man is meant. (Equation 3 and ff)

- If indices of part of a set, then the set should be noted as a set, e.g., $i \in \lbrace1, 2, ..., n\rbrace$ with brackets (in latex they need \ to be displayed).

- "n = [0,1,2,3,4]" is not an array. Please fix this.

- Subsection 3.3 has a typo - the efficiency improvement leads *to* 7 hours computation time in my understanding.

- Can you elaborate why the COVID-19 allocation simulation needs continuous actions? To me it looks it has a big discrete action space, which could be efficient to handle continuously.

- Please streamline lower-upper-casing throughout all plot labels.

- Streamline how to reference figures throughout the paper.

- Please label Figure 5 appropriately. There seems to be something wrong.

- The ax ticks for the middle plot of Figure 6 should be abbreviated.

- Be more specific what is exactly plotted in the plots. Explicit explanation of shaded region etc.

- Please use the decimal separator, e.g., 6,400.

- What is the current state of "Furthermore, we found some mathematical definitions, design choices and surprising results in the original paper somewhat difficult to understand. We sent our questions to the authors, but have not received a response by the time of submission."? As this should be changed before acceptance.

- What I'm still confused about: Is it even possible to do well on different fairness scoring functions at the same time?

- Can you design an experiment that clarifies the impact of counterfactual memories in the COVID-19 environment? And / or elaborate more precisely what impact these have?

**Strengths And Weaknesses:**

The paper is overall well-written and easy to comprehend.

The overall usage of space especially regarding figures leaves room for improvement.

The evidence for the impact of counterfactual memories could be clarified more.

There are multiple weakpoints I tried to address within the Requested Changes section.

---

> ### Author Response · Authors · 2025-03-18
> **Response to your review**
>
> We thank the reviewer for their insightful comments and constructive feedback. We have carefully considered each point of feedback and provide our responses below. Any un-responded-to request has been corrected in the updated manuscript.
>
> - The evidence for the impact of counterfactual memories could be clarified more.
>
> We added the following explanation for the results regarding why an excess of counterfactual memories degrades performance compared to no counterfactual memories.
> “Too many counterfactual memories perform worse than no counterfactual memory augmentation present. This might be due to the increasing proportion of unrealistic counterfactual states that increase with n. With higher values for n, more states that would normally never appear in a trajectory would be produced as counterfactual states, that the DQN learns from. These additional “impossible” counterfactual states add noise to the learning process. For low values of n, the additional benefit of increased (realistic) counterfactual memories seems to outweigh the performance degradation caused by the added noise (of the unrealistic counterfactual memories).”
>
> - Can you elaborate why in Table 1, method RNN, it translates the RNN to a recurrent Q-network? What makes the RNN a Q-network here?
>
> To clarify, the “methods" in Table 1 refer to the way in which memories are stored and accessed. In all cases, a Q-network is used to decide the actions taken, however, each method represents and uses memories differently. In the case of the RNN method, the memories themselves are not explicitly stored but the Q-network network is itself an RNN, which is responsible for storing relevant information from memories (real or counterfactual) and deciding the action. We have clarified the end-to-end nature of this method in the revised submission and made the table headers more explicit.
>
> - Can you elaborate why the COVID-19 allocation simulation needs continuous actions? To me it looks it has a big discrete action space, which could be efficient to handle continuously.
>
> In the revised paper we clarify that the action space is handled more efficiently using a continuous action space rather than being inherent to the problem.
>
> - What is the current state of "Furthermore, we found some mathematical definitions, design choices and surprising results in the original paper somewhat difficult to understand. We sent our questions to the authors, but have not received a response by the time of submission."? As this should be changed before acceptance.
>
> Until now we have not received any response from the authors.
>
> - What I'm still confused about: Is it even possible to do well on different fairness scoring functions at the same time?
>
> When comparing the performance on different scoring functions, we trained our models specifically for each scoring function separately. We didn’t train a single model and evaluate it on different scoring functions at the same time. It is an interesting question whether it is possible for a model to perform well on different scoring functions at the same time, but it is out of the scope of this paper.
>
> - Can you design an experiment that clarifies the impact of counterfactual memories in the COVID-19 environment? And / or elaborate more precisely what impact these have?
>
> We did conduct this experiment, the results are shown in section 4.2.4. We found that model performance decreases when counterfactual memories are used in the COVID-19 environment, and elaborated on our findings in the section.

---

> > ### Comment · Reviewer_pp2W · 2025-04-07
> >
> > Thank you for your answer and for addressing the points raised in my initial review.

---

### Review · Reviewer_4kBh · 2025-03-11

**Summary Of Contributions:**

The paper experimentally analyzes an algorithm FairQCM proposed by previous work to deal with non-Markovian fairness effects i.e., when fairness outcomes of an action depend not only on the current state, but also on previous states. FairQCM is a reinforcement learning algorithm with states including memory beyond the current state and counterfactual memories. The goal of the experiments is to analyze the extent of improvement for fairness and utility when using memory and counterfactual memories.

The paper replicates the results of experiments on the algorithm from previous work and extends the experiments in the following ways: 1) varying the ratio of counterfactual memories to real memories, 2) analyzing two more fairness metrics (egalitarian score and Gini score), 3) considering dynamic parameters in the environment, and 4) testing on a new epidemiology simulation environment.

The paper replicates the original paper's findings. Through their extended experiments, they arrive at the following findings:
- In the environments of past work, both real and counterfactual memories are beneficial. However, counterfactual memories are only beneficial up to a certain extent. Once the counterfactual memories exceed a certain ratio of the real memories, the outcome is worse than having no counterfactual memories at all.
- FairQCM also performs well relative to the egalitarian fairness metric. However, it has lower fairness and utility when the fairness metric is Gini welfare score.
- FairQCM offers more clear benefits when the parameters of the environment are dynamically changing.
- In the epidemiology simulation environment, counterfactual memories have a negative impact. They hypothesize that this is due to noise in the approximation of counterfactual memories in the more complex environment of the epidemiology simulation.

**Audience:**

Yes

**Claims And Evidence:**

Yes

**Requested Changes:**

I think the paper needs more discussion of the observed results to secure my acceptance recommendation. Especially why some of the findings from previous work differ from those in the new environment and other fairness scores.

1. What is it about the Gini score that makes FairQCM perform worse than other baselines?
2. Why do you think having too many counterfactual memories is worse than having no counterfactual memories at all?
2. To support your hypothesis that the noise in approximating counterfactual memories is the reason why using counterfactual memories has a negative effect in the epidemiology simulation, can you show how varying the accuracy (through varying the approximating model's parameters) changes the outcomes?

**Strengths And Weaknesses:**

Strengths:
1. Extending the experiments to more complex and realistic environments inspired by real-life applications highlights the impacts of various choices in FairQCM in realistic scenarios.
2. Demonstrating that different baselines are better for other fairness metrics such as gini score shows how the choice of method should rely on the fairness metric of interest.
3. More fine-grained analysis of the impact of real and counterfactual memories.
4. Reproducible code for the experiments.

Weaknesses
1. Lack of novelty in methods or experiments
2. Lack of analysis and discussion of the observed results

---

> ### Author Response · Authors · 2025-03-18
> **Response to your review**
>
> Thank you for taking the time to review our work. We appreciate the reviewer’s detailed feedback on our submission and we address their requested changes in our response below.
>
> - What is it about the Gini score that makes FairQCM perform worse than other baselines?
>
> The primary reason for testing different fairness functions was to investigate the robustness of FairQCM across them. Why exactly the Gini score makes FairQCM perform worse we cannot explain. Additionally, we believe that further experiments to explain this drop in performance when using Gini would be out of scope of this paper but would be an interesting subject for future research.
>
> - Why do you think having too many counterfactual memories is worse than having no counterfactual memories at all?
>
> We added our answer and reasoning to this question to the updated version of the paper:
>
> Too many counterfactual memories perform worse than no counterfactual memory augmentation present. This might be due to the increasing proportion of unrealistic counterfactual states that increase with n. With higher values for n, more states that would normally never appear in a trajectory would be produced as counterfactual states, that the DQN learns from. These additional “impossible” counterfactual states add noise to the learning process. For low values of n, the additional benefit of increased (realistic) counterfactual memories seems to outweigh the performance degradation caused by the added noise (of the unrealistic counterfactual memories).”
>
> - To support your hypothesis that the noise in approximating counterfactual memories is the reason why using counterfactual memories has a negative effect in the epidemiology simulation, can you show how varying the accuracy (through varying the approximating model's parameters) changes the outcomes?
>
> In this context, “noise” refers to the slightly perturbed environment dynamics the model learns due to the added counterfactual memories (we clarify this in section 4.2.4). We believe that varying the model’s accuracy (for example by adding numerical perturbations to its parameters) would be unrelated to this effect, as this would make the model’s parameters noisy, and not the dynamics of the environment.

---

> > ### Comment · Reviewer_4kBh · 2025-04-07
> >
> > Thank you for your response addressing my questions

---

### Decision · Action_Editor_s1bw · 2025-04-27

**Recommendation:** Accept with minor revision

**Comment:**

One reviewer raised a question about the methodology and contributions around the different scoring functions.  The authors responded that:

When comparing the performance on different scoring functions, we trained our models specifically for each scoring function separately. We didn’t train a single model and evaluate it on different scoring functions at the same time. It is an interesting question whether it is possible for a model to perform well on different scoring functions at the same time, but it is out of the scope of this paper.

This makes sense, but it would be good to incorporate this into the paper itself.  In the latest version it mentions training separate models from scratch for each in 3.2.2, but the results in 4.2.2 just talk about FairQCM rather than being more precise about which version in each instance and the fact that this question isn't explored should be explicitly noted.

With these small changes to 4.2.2 I believe all concerns will have been addressed.

**Audience:**

All reviewers agree that this standard is met.  The paper deepens our understanding of FairQCM and introduces a novel environment motivated by COVID-19.  Both of these are likely to be of interest to some members of TMLR's audience.

**Claims And Evidence:**

All reviewers agree that this standard is met.  The paper is clear about its goals and contributions and appropriately positioned relative to prior work.  The reviewers raised a number of small concerns or places where greater clarity was needed, and with one exception below these appear to be addressed.

---

> ### Author Response · Authors · 2025-05-06
> **Response to decision**
>
> Dear Action Editor and Reviewers,
>
> Thank you for your helpful suggestions. We have revised the paper accordingly based on your feedback and incorporated the proposed changes to section 4.2.2.
>
> We are grateful for your time, thoughtful feedback, and constructive guidance, which have greatly contributed to improving the quality of our submission.
>
> Kind regards,
>
> Authors